

# Modelling stream flow with a discrete rainfall-runoff model and 37GHz PDBT microwave observations: the Xiangjiang River basin case study

Haolu Shang[1,2], Massimo Menenti[2,1], Li Jia[1]

[1]Stake Key Laboratory of Remote Sensing Science, Institute of Remote Sensing and Digital Earth, Chinese Academy of Sciences. Beijing 100101, China
[2]Geoscience and Remote Sensing, Delft University of Technology, Delft, 2628 CN Delft / 2600 GA Delft, The Netherlands

*Correspondence to*: Li Jia (jiali@radi.ac.cn)

**Abstract.** A discrete rainfall-runoff model has been developed, which uses retrievals of Water Saturated Soil (WSS) and inundation area from 37GHz microwave observations. The model was implemented at three levels of increasing complexity using field-measured ground water table, WSS and inundated area, and precipitation data. The three levels, defined by the key-variables are: 1) precipitation and base flow; 2) overland flow, infiltrated flow and base flow; 3) overland flow, potential subsurface flow and base flow. The base flow is estimated from observed ground water table depth, while overland and infiltrated flows are estimated from precipitation and the WSS and inundated area. A linear scaling method is developed to estimate the potential subsurface flow. The three model implementations are calibrated with the gauge measurements of 10-day average river discharge in 2002 and 2005 respectively at Changsha station, downstream of Xiangjiang River basin, China. The discrete rainfall-runoff model assumes that specific runoff is determined by antecedent precipitations over a variable period of time. This duration is a model parameter varying between 10 and 150 days. The performance of the discrete rainfall-runoff model increased with the duration of antecedent precipitation for all three implementations in both years. With a duration of 150 days, the model reaches its best performance: Nash-Sutcliffe Efficiency, NSE, for the 1st implementation was ≥ 0.90 with relative RMSE ≤ 22 %; NSE ≈ 0.99 with relative RMSE ≤ 5 % for the 2nd implementation, and NSE ≥ 0.99 with relative RMSE ≤ 4 % for the 3rd one. These good performances prove that the retrievals of WSS and inundated area clearly improve model accuracy, thus justifying the choices of parameters and the method to estimate the potential subsurface flow. The set of parameters driving each implementation is an indication of dominant hydrological processes, particularly water storage, in determining the catchment response to rainfall. Significant differences in the annual water yield have been observed across the three implementations. The relative RMSE in each season demonstrates the possible recharge period of the ground water in Xiangjiang River basin.

Keywords: Passive microwave, Water Saturated Soil, Discrete rainfall-runoff model, Regional water storage.



## 1 Introduction

Due to the sensitivity to water content in the top soil layer, passive microwave observations at 37GHz have been used to observe water saturated soil and inundated area (Choudhury, 1989, 1991; Giddings and Choudhury, 1989; Hamilton et al., 2002, 1996; Shang et al., 2014; Sippel et al., 1994; Sippel et al., 1998; Temimi et al., 2005). The inundated area within a river basin can be related to river discharge directly, thus 37GHz microwave observation have been used to estimate river discharges using empirical relationships (Brakenridge et al., 2007; Vörösmarty et al., 1996). We have explored a further development based on these studies by modelling stream flow using the microwave observations at 37 GHz directly to parameterize the rainfall-runoff relationship under changing climate and increasing anthropogenic impacts.

## Review on WSS and inundated area and its application problems in hydrology models

The Water Saturated Soil (WSS) and inundated area, retrieved from Polarization Difference Brightness Temperature (PDBT) at 37GHz(Shang et al., 2014), is where open water or water-saturated top soil occurs, such as wet lands, ponds, lakes and rivers. Several studies indicate that the WSS and inundated area is a determinant of stream flow. As documented by Miller and Nudds (1996), the changes of wetland area in floodplain significantly influence changes of stream flow in the Mississippi River Valley. In our previous study (Shang et al., 2014), the upstream WSS and inundated area were highly correlated with the area of the Poyang Lake downstream. In some conceptual hydrology models, such as Xinganjiang model (Zhao, 1977, 1984), PDM rainfall-runoff model (Moore, 2007, 1985), TOPMODEL (Beven and Kirkby, 1979; Beven et al., 1984), VIC (Liang et al., 1994; Wood et al., 1992) and ARNO (Todini, 1996) , the saturated soil area is a conceptual parameter which determines the fast runoff produced after a storm, i.e. overland flow and fast subsurface flow. In a catchment, the depth of saturated soil area varies with topography and water storage (Beven and Kirkby, 1979; Wood et al., 1992), while the WSS and inundated area retrieved with the 37GHz observations is a surface feature. This difference implies that the retrievals of WSS and inundated area cannot be directly used in these hydrological models.

In hydrological models, the saturated soil area is an element of regional water storage. The regional water storage capacity is estimated from physically based model parameters, such as the topography-soil index in TOPMODEL (Sivapalan et al., 1987) or the infiltration factor in VIC (Liang et al., 1994; Wood et al., 1992). These parameters determine the saturated soil area when a storm occurs. Thus a possible solution is to estimate the regional water storage from the retrievals of the WSS and inundated area. This is similar to soil moisture estimation done with 37GHz microwave observation (Lakshmi et al., 1997a, b). The studies of wetlands in the prairies of North America (Hayashi et al., 1998; Kazezyılmaz-Alhan et al., 2007; Shjeflo, 1968; Su et al., 2000; Winter and Rosenberry, 1995; Winter and Valk, 1989; Woo and Rowsell, 1993) show, however, that the hydrological processes in wetlands are very complicated and their description requires many spatially detailed data that are difficult to access in most river basins. Particularly, it is difficult to estimate regional water storage from surface wetness, thus making the retrievals of the WSS and inundated area very useful.





**Discrete form of the water balance model**

The WSS and inundated area retrieved from 37GHz PDBT describe the surface wetness conditions of a catchment. We developed a new rainfall-runoff model to directly use this information, based on the water balance model in the Budyko framework (Budyko, 1971; Donohue et al., 2007). The WSS and inundated area are used to describe the generation of stream

flow in a similar way as in TOPMODEL (Beven and Kirkby, 1979; Sivapalan et al., 1987) and VIC (Liang et al., 1994; Wood et al., 1992).

In the Budyko framework, the change in the regional soil water storage is the difference between precipitation and outflows, i.e. stream flow and evapotranspiration, during a certain period. The period needs to be longer than the time scale of fluctuations in soil water storage, thus the model describes the quasi-steady-state conditions as in Donohue et al. (2007);

Eagleson (1978). If we could explicitly measure all the terms, including aquifer water storage, of the water balance within each time interval, the discrete form of the water balance model would be:

$$\sum_{t=0}^{N} \Delta S_t + \sum_{t=0}^{N} \Delta G_t + G_m = \sum_{t=0}^{N} P_t - \sum_{t=0}^{N} Q_t - \sum_{t=0}^{N} E_t + Q_{min}^b \quad (1)$$

Where $\Delta S_t$, $\Delta G_t$, $P_t$, $Q_t$, and $E_t$ is respectively changes in soil water storage, changes in aquifer water storage, precipitation, stream flow and evapotranspiration in the $t^{th}$ interval. The time interval can be daily, weekly, or monthly. N is the total

number of time intervals, i.e. the temporal coverage of the analysis (Donohue et al., 2007). At $t = 0$, soil water storage, precipitation and evapotranspiration are all assumed to be zero, and only the minimum aquifer water storage, i.e. $G_m$ in the l.h.s. of the Eq. (1), yields the minimum base flow, i.e. $Q_{min}^b$ in the r.h.s. of the Eq. (1). $G_m$ and $Q_{min}^b$ are assumed equal to each other, thus they cancel out in Eq. (1). Since we are interested in the stream flow at the $N^{th}$ time step, Eq. (1) can be reorganized as:

$\quad Q_N = w_N \times P_N + \sum_{t=0}^{N-1} w_t \times P_t \quad (2)$

where

$w_N = (P_N - \Delta S_N - \Delta G_N - E_N)/P_N$

$w_t = (P_t - \Delta S_t - \Delta G_t - E_t - Q_t)/P_t$

Equation (2) means that the stream flow can be estimated as the weighted sum of precipitations in the current and antecedent

time steps. For each antecedent time step, the remaining water balance terms do not balance, i.e. $w_t \neq 0$ in Eq. (2), because a catchment takes time to respond to a storm event. For example, in a watershed the time-lag between a storm and runoff is large and variable (Gleick, 1987), and the discharge and recharge rates between soil layers and aquifers are different, which makes it a challenge to estimate water storage as a function of time (Eagleson, 1978).

To close the water balance as in Eq. (2), conceptual hydrological models, such as VIC and ARNO, allocate precipitation to

the different terms in each time interval, where stream flow is replaced with direct runoff. These different terms yield different components of stream flow. The problem is that the interaction between storage terms, i.e. water in soil and aquifer, and stream flow also occurs in a catchment (Mertes, 1997). To consider this dynamic redistribution of rainfall in a certain time, we have rewritten Eq. (2) to derive a discrete rainfall-runoff model:



$$Q_N = \sum_{i=0}^{N} w_i \times P_i \quad (3)$$

where

$$w_i = \left( P_i - \sum_{j=i}^{N} \Delta S_i^j - \sum_{j=i}^{N} \Delta G_i^j - \sum_{j=i}^{N} E_i^j - \sum_{j=i}^{N-1} Q_i^j \right) \Big/ P_i$$

where $P_i$ is the storm rainfall at the $i^{th}$ time step and $w_i$ is the contribution weight of $P_i$ to current stream flow, i.e. $Q_N$. The $i^{th}$ time step precedes the $N^{th}$ time step, thus the time required to redistribute precipitation is $N\text{-}i\text{+}1$. $\sum_{j=i}^{N} \Delta S_i^j$, $\sum_{j=i}^{N} \Delta G_i^j$ and

5 $\sum_{j=i}^{N} E_i^j$ is $P_i$ stored into soil, aquifers, and consumed by evapotranspiration, respectively, in the period of time $N\text{-}i\text{+}1$. $\sum_{j=i}^{N-1} Q_i^j$ is the stream flow generated by $P_i$ in the period of time $N\text{-}i$. When $i = N$, $\sum_{j=i}^{N-1} Q_i^j$ is equal to 0. For each $P_i$, the weight $w_i$ is a function of $N$. This means $P_i$ is re-distributed into soil water, aquifer water, evaporation, transpiration, stream flow in a catchment during the period of time $N\text{-}i\text{+}1$. Theoretically, the weights in Eq. (3) can be determined by calibrating this conceptual model with observed stream flows and precipitations by applying linear recursive regression analysis

(Young, 1984).

Time series of remote sensing observations of precipitation (e.g. by TRMM), surface wetness condition (i.e. WSS and inundated area) and even stream flow in the future, i.e. by SWOT (Bates et al., 2014; Gleason and Smith, 2014; Paiva et al., 2015), can be applied to determine the weights $w_i$ using Eq. (3). Another advantage of using Eq. (3) is that $w_i$ can be easily adapted through model calibration to capture changes in the catchment response to rainfall. The catchment response, for

example, changes between dry and wet years (Dunne, 1978). Anthropogenic modifications in land use and land cover have a large impact on catchment response (Barnett et al., 2008; Jaramillo and Destouni, 2015). Especially in China, previous studies show that human impacts account for above 40 % of total annual stream flow changes in recent years, as the examples shown in the Northern China (Chang et al., 2015; Li et al., 2007).

**The three implementations of the discrete hydrology model**

The WSS and inundated area describe surface wetness, while ground water table depth describes the wetness conditions at the lower boundary of the soil-vegetation system in a catchment. These two boundary conditions can help us to simplify the calibration of the discrete rainfall-runoff model (Eq. (3)) and improve the model performance. Precipitation is redistributed into different component flows, which reach river channels with different time lags (Beven and Kirkby, 1979; Moore, 1985;

Sivapalan et al., 1987; Todini, 1996; Wood et al., 1992; Xu and Singh, 2004). Assuming model calibration is accurate, including the estimation of time-lags, the precipitations in Eq. (3) can be replaced with different sub-set of component flows. We proposed below three different conceptual discrete rainfall-runoff models, based on different choices of dominant component flows.





First implementation. To use Eq. (3), we assume that the minimum regional storage is equal to $G_{min}$. This assumption, however, is questionable in many river basins, and the timing of minimum regional water storage is difficult to estimate. A realistic solution to this problem is to assume that precipitation, which is older than a certain antecedent time step, has been stored in aquifers as ground water. The ground water produces base flow as a linear reservoir. Thus only the precipitation

events in certain antecedent period of time contribute to the stream flow as described by Eq. (3), i.e. $i > 0$ in the Eq. (3). This duration of antecedent precipitation is a critical parameter in the three implementations of the discrete rainfall-runoff model and needs to be set prior to model calibration. By introducing the base flow and the duration of antecedent precipitation, the discrete rainfall-runoff model can be simplified and calibration is easier (see Sect. 2.1 for details). We will also illustrate how the model performance changes with various durations of antecedent precipitation for all three implementations of the

discrete rainfall-runoff model.

Second implementation. Besides base flow, we can partition the precipitation at each time step into overland and infiltrated flow, taking surface wetness conditions into account, i.e. by using the retrievals of the WSS and inundated area. This implementation is based on the water balance at the surface. The stream flow is the sum of three components: the

accumulated weighted sum of overland flows, the accumulated weighted sum of infiltrated flows and the base flow.

Third implementation. A fraction of the infiltrated flow will result in subsurface flow when it encounters saturated soil layers underneath the soil surface (Beven and Kirkby, 1979; Beven et al., 1984; Sivapalan et al., 1987), which we define as potential subsurface flow. We assume that the potential subsurface flow scales linearly with ground water table depth. In the

third implementation, the potential subsurface flow replaces the infiltrated flow in the second implementation. This assumption will be evaluated by comparing the model performance in the third implementation with that in the second implementation.

In the second and third implementations of the discrete rainfall-runoff model, the weight of each component flow quantifies

which fraction of this component flow contributes to the stream flow at the current time step. The precipitation weight at each antecedent time step in the second and third implementations can be calculated by adding up the weights of the component flows.

Overall, the objective of this study is to develop a new rainfall-runoff model to use directly the retrievals of the WSS and inundated area from 37GHz microwave observations. The model is derived from the water balance equation, but fully

considers the dynamic distribution of precipitation in a catchment. The term "discrete rainfall-runoff model" relates to the use of discrete time steps of antecedent precipitation to estimate current stream flow. In the three implementations of the discrete rainfall-runoff model, the WSS and inundated area is used to estimate component flows produced from discrete precipitation events, and the field measurements of ground water table depth are used to estimate the base flow. The method applied to  retrieve the WSS and inundated area from 37GHz PDBT has been described in detail by Shang et al. (2014).



There are five sections in this paper: Introduction, Methods, Data and Study Area, Results and Discussion and Conclusion. In the Method section, the three implementations of the discrete rainfall-runoff model are explained in detail. We applied our models to the middle and upstream reach of Xiangjiang River basin, in a dry and a wet year. A short description of the river basin and of model input data is provided in the section of Dataset and Study Area. In the Result and Discussion section, we

5   describe numerical experiments on stream flow with various durations of antecedent precipitation to illustrate how the model performance is improved by increasing the duration of antecedent precipitation. We also interpret the temporal patterns in the weights and evaluate the method to estimate the potential subsurface flow. Clear differences were observed in weight values on dry and wet years. Model validation is also demonstrated.

## 2. Method

The calibration of the discrete rainfall-runoff model developed above, i.e. Eq. (3), is a problem, since it is difficult to estimate precisely when minimum regional water storage occurs. We will explain how to solve this problem and introduce the three implementation forms of the discrete rainfall-runoff model in sequence. The first implementation of the discrete hydrological model is derived (Sect. 2.1) by modifying Eq. (3) to account for base flow produced from ground water.

According to the water balance at the surface, the WSS and inundated area are used to partition the precipitation into overland and infiltrated flows. The second implementation of the discrete hydrology model replaces the precipitation in the first implementation with the overland flow and infiltrated flow. A method is developed to estimate the potential subsurface flow from the infiltrated flow as a function of ground water table depth. The potential subsurface flow replaces the infiltrated flow in the second implementation, to derive the third implementation of the discrete hydrology model (Sect. 2.3). These

three implementations of the discrete hydrology model have increasing complexity. The calibration method and criteria to evaluate model performance are introduced within reference to the 1[st] implementation (Sect. 2.1), then applied to the other two implementations. The retrievals of the WSS and inundated area from 37GHz PDBT data have been described in detail by (Shang et al., 2014).

**2.1 The first implementation of the discrete rainfall-runoff model: model calibration and performance**

To calibrate the model defined by Eq. (3), i.e. to determine the weight $w_i$, we partition the stream flow into different component flows as done by Xu and Singh (2004). We assume that there is an antecedent time step, i.e. $m$ in Eq. (4), before which all precipitation in a catchment is used to recharge the ground water reservoir and to produce base flow, thus Eq. (3) can be rewritten as:

$Q_n = \sum_{i=n-m}^{n} w_i \times P_i + Q_n^b$   (4)

where





$$Q_n^b = \sum_{i=0}^{n-m-1} w_i \times P_i = k \times G_n + B$$

$Q_n^b$ is the base flow produced from the ground water reservoir $G_n$ at $n^{th}$ time step as a linear reservoir with a releasing factor, i.e. $k$, plus a constant flow, i.e. B. The ground water reservoir is described by the ground water table depth. The duration from the $m^{th}$ to $n^{th}$ time step is the duration of antecedent precipitation. The weights of precipitation events in Eq. (4), i.e. $w_i$, have the same meaning as those in Eq. (3).

It is important to mention that the ground water table depth used in Eq. (4) may not be a spatial variable. The releasing factor and constant flow, i.e. $k$ and $B$, will be calibrated against observed stream flow, i.e. for the entire catchment, in order to balance the ground water discharge due to the precipitation events involved in Eq. (4) with the discharge due to even older precipitation events. This also indicates that a limited number of field measurements of ground water table is sufficient to estimate the parameters in Eq. (4), since the fluctuations of regional ground water table depth, rather than its spatial pattern, have to be reproduced.

**Model calibration**

The duration of antecedent precipitation events in Eq. (4) is assumed to be constant at each time step within one year, i.e. m is constant in Eq. (4). Once "m" is set, the structure of Eq. (4) is fixed and the model parameters, i.e. $w_i$, $k$ and $B$, can be estimated by model calibration against observed stream flow using time series of observed precipitation and ground water table depth. The value of "m" needs to be less than the total number of samples in the annual time series, i.e. "n". The linear least squares method is used to solve such over-determined linear equation, so that the model parameters minimize the difference between observed and modelled stream flows. The optimal model parameters in Eq. (4) are determined by solving the equation:

$$(\mathbf{X^T X})\beta = \mathbf{X^T Y} \qquad (5)$$

where

$$\mathbf{X} = \begin{bmatrix} P_0 & P_{-1} & \cdots & P_{-m} & G_1 & 1 \\ P_1 & P_0 & \cdots & P_{1-m} & G_2 & 1 \\ \vdots & \vdots & \ddots & \vdots & \vdots & \vdots \\ P_n & P_{n-1} & \cdots & P_{n-m} & G_n & 1 \end{bmatrix}, \beta = \begin{bmatrix} w_n \\ w_{n-1} \\ \vdots \\ w_{n-m} \\ k \\ B \end{bmatrix}, \text{ and } \mathbf{Y} = \begin{bmatrix} Q_0 \\ Q_1 \\ \vdots \\ Q_n \end{bmatrix}$$

In Eq. (5), the precipitations in $\mathbf{X}$ include the observations in the antecedent year, i.e. [P$_{-1}$, P$_{-2}$, …, P$_{-m}$]. The derivation of Equation (5) and more detailed information on the linear least squares method can be found in many handbooks, such as Björck (1996); Lawson and Hanson (1974).

In the matrix $\mathbf{X}$, the precipitation at $i^{th}$ time step exists "m+1" times, i.e. from $i$ to $i+m$ rows, which means precipitation will contribute to the stream flow until the $(m+1)^{th}$ time step. So we call "m" the duration of antecedent precipitation in a catchment. The column position of the $i^{th}$ precipitation ranges from 1 to m+1 in $\mathbf{X}$. This means the weight parameters, i.e. [$w_n, w_{n-1}, …, w_{n-m}$] in Eq. (5), also represent the evolution of the contribution of the $i^{th}$ precipitation events to stream flow at





various time steps during the duration of antecedent precipitation. The same method applies to the second and third implementations of the discrete rainfall-runoff model.

Thus there are two ways to interpret the weight parameters of precipitation: 1) in relation with the evolution of stream flow; 2) in relation with the evolution of the precipitation contribution. For stream flow at each time step, i.e. the elements in Matrix **Y** in Eq. (5), the precipitation events that contribute to the stream flow, i.e. the elements in each row of Matrix **X**, change with the reference time step. The fraction of stream flow due to each precipitation event is determined by both the weight and the amount of each precipitation. Thus it will be complicated to interpret the weight pattern in relation with stream flow at different time steps. In the Results and Discussion section we propose to interpret the pattern of weights in relation with the evolution of precipitation contributions to stream flow at a given antecedent duration.

**Model performance**

To evaluate the model performance, two criteria were used in this study: 1. the Nash-Sutcliffe Efficiency (Nash and Sutcliffe, 1970) , i.e. NSE:

$$\text{NSE} = 1 - \sum_{i=1}^{N}(E_i - O_i)^2 / \sum_{i=1}^{N}(O_i - \bar{O})^2 \quad (6)$$

where $E_i$ and $O_i$ is the model estimated and observed stream flow at $i^{th}$ time step respectively, $\bar{O}$ is the average of measured stream flows, N is the total number of observations;

and 2. Relative RMSE (RRMSE):

$$\text{RRMSE} = 100\% \times \sqrt{\sum_{i=1}^{N}(E_i - O_i)^2 / N} \Big/ \bar{O} \quad (7)$$

NSE indicates how well the model estimates replicate the observations, while RRMSE is a measure of the bias in the model estimates.

**2.2 The second implementation form of the discrete rainfall-runoff model**

When precipitation reaches the surface, it is partitioned into the overland and infiltrated flow, depending on surface wetness, e.g. as parameterized by the WSS and inundated area (Beven and Kirkby, 1979; Liang et al., 1994; Moore, 1985; Sivapalan et al., 1987; Todini, 1996).  Thus the precipitation in Eq. (4) can be replaced by the two component flows with their independent weights as follows:

$$Q_n = \sum_{i=n-m}^{n} \beta_i^{21} \times Q_i^O + \sum_{i=n-m}^{n} \beta_i^{22} \times Q_i^I + Q_n^b \quad (8)$$

where

$$Q_i^O = WS_i \times P_i$$

$$Q_i^I = (1 - WS_i) \times P_i$$

$\beta_i^{21}$ and $\beta_i^{22}$ is the weight of overland flow (i.e. $Q_i^O$) and infiltrated flow (i.e. $Q_i^I$)) at the $i^{th}$ antecedent time step, respectively; $Q_i^O$ and $Q_i^I$ are the fractions of precipitation (i.e. $P_i$), determined by the fraction of Water Saturated Soil (WSS)





and inundated area (i.e. $WS_i$). In Eq. (8), we assume that:1) evapotranspiration is a fraction of both $Q_i^O$ and $Q_i^I$; 2) subsurface flow is produced by $Q_i^I$. So both $Q_i^O$ and $Q_i^I$ result from the instantaneous water balance at the surface. We did not take interception by the vegetation canopy into account in Eq. (8).

A very important reason to use the fraction of the WSS and inundated area in Eq. (8), is that $\beta_i^{21}$ and $\beta_i^{22}$ at the same $i^{th}$

antecedent time step may not be equal to each other. The overland water flows over the surface to channels and rivers, while the infiltrated water percolates through soil layers first and then runs through soil layers to channels and rivers. The water flow velocity is different on the surface and in the soil. Anthropogenic changes in land cover mainly modify overland and infiltrated flows, which will also change the temporal pattern of $\beta_i^{21}$ and $\beta_i^{22}$. Accordingly, in the second implementation of our model, the weights of precipitation, i.e. $w_i$ in Eq. (4), can be calculated as:

$w_i = \beta_i^{21} \times WS_i + \beta_i^{22} \times (1 - WS_i)$    (9)

The physical meaning of precipitation weights in Eq. (9) is the same as in Eq. (3) and Eq. (4), but the use of the retrieved $WS_i$ in Eq. (9) makes it easier to estimate the $w_i$.

**2.3 The third implementation form of the discrete rainfall-runoff model**

The infiltrated flow in Eq. (8) can be further partitioned into two components: one part is stored into soil layers and aquifers, and the rest becomes subsurface flow when it reaches the saturated soil layer underneath the surface. It is difficult to estimate the fraction of infiltrated flow converted into subsurface flow, thus we estimate the potential subsurface flow and replace the infiltrated flow in Eq. (8) as follows:

$Q_n = \sum_{i=n-m}^{n} \beta_i^{31} \times Q_i^O + \sum_{i=n-m}^{n} \beta_i^{32} \times Q_i^S + Q_n^b$    (10)

where

$Q_i^S = Q_i^I \times (G_i - G_{min})/(G_{max} - G_{min})$

$\beta_i^{31}$ and $\beta_i^{32}$ are the weights of overland flow and potential subsurface flow (i.e. $Q_i^S$) at $i^{th}$ time step respectively; $Q_i^S$ is potential subsurface flow, which indicates the potential proportion of the infiltrated flow transformed into subsurface flow. $Q_i^S$ is estimated using observations of the ground water table depth (i.e. $G_i$) scaled by the range between maximum and

minimum ground water table depth in each year (i.e. $G_{max}$ and $G_{min}$).

We assume that the depth of the saturated soil layer is equal to the ground water table depth. When the ground water table depth increases, the residual storage capacity in soil layers decreases. This means that a larger fraction of infiltrated water becomes subsurface flow and flows into river channels. Whether the potential subsurface flow can be estimated using the simple Eq. (10) is still a question, thus we will compare $\beta_i^{31}$ with $\beta_i^{21}$, and $\beta_i^{32}$ with $\beta_i^{22}$ respectively in the section Results

and Discussion. The precipitation weights in the third implementation of the discrete rainfall-runoff model can then be calculated as:

$w_i = \beta_i^{21} \times WS_i + \beta_i^{22} \times (1 - WS_i) \times (G_i - G_{min})/(G_{max} - G_{min})$    (11)





In Eq. (11), the surface wetness conditions (i.e. $WS_i$) and the ground water table depth (i.e. $G_i$) are the boundary conditions of water flow in the soil. It will be very interesting to evaluate whether the boundary conditions of a complicated system (i.e. soil layer) can be used to model the outflow produced from this system (i.e. overland flow and subsurface flow), so that the complicated flow processes, i.e. water flow in vadose zone and evapotranspiration, can be described in a simpler way.

## 3. Data and Study Area

The study area is the Changsha upstream catchment in the Xiangjiang River basin (blue polygon in Fig. 1). The study area covers 88,125 km$^2$, located at 25°12'32.29" N - 28°29'30.11" N and 111°01'10.29" E - 113°52'58.29" E. Annual

precipitation in this region is on average 1700 mm. The rainy season is from April to June. Abundant water resources make the Xiangjiang River basin as one of the best irrigated floodplains in China. Paddy fields consume the largest share of water resources. Many small lakes, wetlands and fish ponds are the natural reservoirs of the river basin. There are also many artificial reservoirs distributed along the tributaries and main stream of the Xiangjian River, in order to meet increasing requirements for electrical energy from industry and cities.

We used observations, i.e. 9 wells in Changsha, of the ground water table depth averaged over 10-days. All the other input data were also averaged over 10-days. The input remote sensing data was 10-day averages of the WSS and inundated area data (Fig. 2) over the whole study area, which is retrieved from daily 37GHz PDBT observations as shown in Shang et al. (2014). We need to mention that in September 2005, the extremely high WSS and inundated area is probably due to the Typhoon Talim occurred during this period. The 10-day averages of precipitation data (Fig. 2) was extracted from ITP-

atmosphere forcing data set (Chen et al., 2011; Yang et al., 2010) over the whole study area. This data set is based on calibration of TRMM precipitation retrievals against rain gauges in China, thus we believed our precipitation data to be better than the original TRMM retrievals. The daily stream flow at the Changsha hydrology station is not affected by water consumed for irrigation and by water storage in the reservoir. The daily unimpaired stream flow was also averaged over 10-day (Fig. 2) and used to calibrate the discrete rainfall-runoff model. The calibration period was in 2002 (wet year) and 2005

(dry year). The annual precipitation in 2005 was less than in 2002.

## 4. Results and Discussion

### 4.1 General

We developed the three implementations of the discrete rainfall-runoff model with increasing complexity: The first

implementation with precipitation and base flow; the second one with overland flow, infiltrated flow and base flow; the third



one with overland flow, potential subsurface flow and base flow. They are applied to estimate the 10-day averaged stream flow at Changsha station in 2002 (wet year) and 2005 (dry year) respectively. All the inputs are 10-day averages and at the same time step, i.e. time interval is 10 days. The duration of antecedent precipitation was increased stepwise from 1 to 15 antecedent time steps, in order to simulate stream flow and demonstrate how the model performance changes with increased durations. The RRMSE in each season was calculated to illustrate how the model bias changes with the increasing duration. By analysing the changes in seasonal model bias for the second and third implementations, we can estimate timing of ground water recharge periods (see section 4.3).

The stream flow at a given step depends on precipitation and its component flows, but this relationship changes in time. On the other hand, the weight parameter pattern illustrates the evolution of the precipitation contribution to stream flow at a given antecedent time step, i.e. the evolution of $P_i$ in the matrix X in Eq. (5). Thus the meaning of the weight parameter pattern in each implementation is explained in relation with precipitation events or with component flow contributions to stream flows at a given antecedent time step. The comparison of weight parameter patterns between dry and wet years illustrates different catchment responses under various wetness conditions (see Sect. 4.2 and Sect. 4.3). The performance of the 3$^{rd}$ implementation is used to evaluate the method to estimate the potential subsurface flow (Sect. 4.4). We will also compare the weight parameter patterns between the second and third implementations in Sect. 4.4. For each implementation, the difference of $k$ and $B$ in various durations is very small, thus they are not illustrated specifically in this case.

We also evaluate the model parameters of each implementation with the observed stream flow in 2001 (Sect. 4.5). The annual average precipitation of 2001 is close to 2002, thus it is a relatively wet year. For the validation of each implementation, the model parameters, in terms of the weights and the duration of antecedent precipitation, are derived from three experiments: 1) calibration with observed stream flow in 2002: 2) calibration with observed stream flow in 2005; 3) parameter averages over 2002 and 2005.

## 4.2 Simulation with the first implementation of the discrete rainfall-runoff model

In the first implementation of the discrete rainfall-runoff model, the stream flow is simulated with the duration of antecedent precipitation increasing from 1 to 15 time steps (i.e. from 10 days to 150 days) in both 2002 (wet year) and 2005 (dry year). As the three examples of simulated stream flows show (Fig. 3), the estimated bias decreased in both years with increasing duration of antecedent precipitation. For example, before September 2002, the stream flow with the duration of 1 time step is always lower than the observations (Fig. 3a), while later on, the model estimates are higher than the observations. By increasing the duration to 8 and 15 time steps, the bias decreased (Fig. 3a). This improvement indicates that the stream flow production before September 2002 is influenced not only by the recent precipitations but also by precipitation over a longer antecedent period. In 2005 (Fig. 3b), however, the situation is different, suggesting a different catchment response. After 2002, the annual precipitation (from ITP forcing data) shows a decreasing trend. The continuous dry conditions from 2003 to 2005 accelerated the water cycle and shortened the duration of antecedent precipitation events involved into stream flow



production in this catchment. Thus with the same duration of antecedent precipitation (as shown in Fig. 3b, and comparing the model performance in 2002 (Fig. 6) with 2005 (Fig. 7)), the first implementation of the discrete rainfall-runoff model fits the observations better in 2005 than in 2002.

The influence of the duration of antecedent precipitation on model performance can be clearly illustrated by the results obtained with various durations (Fig. 6 and Fig. 7). No matter how the catchment response changes between dry and wet years, with the duration of antecedent precipitation increasing from 1 to 15 time steps, the *NSE* of the first implementation increases from 0.78 to 0.91 in 2002 (Fig. 6a) and from 0.82 to 0.94 in 2005 (Fig. 7a), while the RRMSE decreases from 34 % to 22 % in 2002 (Fig. 6b) and from 28 % to 17 % in 2005 (Fig. 7b). This proves our assumption that the catchment transforms a fraction of precipitations at each antecedent time step to the current stream flow. We also calculated the RRMSE for the first implementation in each season, taking both 2002 and 2005 into consideration (Fig. 8a). With increasing durations, the RRMSE after June decreases. This indicates that precipitation in spring, significantly influences the stream flow after June. This is consistent with other studies on the floods in the Yangtze River basin (Lin and Lu, 1991; Zhou, 1991; Zong and Chen, 2000), where the magnitude of floods after June is related to the rain in spring .

By adding the base flow to the discrete rainfall-runoff model (Eq. (4)), the contribution weights of precipitation in Eq. (3) can be calculated with various durations (Fig. 9) for each year. The weight pattern, in terms of the timing of maximum weight, is similar with different antecedent durations for each year. On the other hand, a significant difference in weight pattern can be found between 2002 (Fig. 9a) and 2005 (Fig. 9b): in 2002 (wet year), the higher weights occur at antecedent time steps 1 and 2, i.e. 10 and 20 days ahead of the current stream flow; while in 2005(dry year), the highest weight occurs at longer antecedent time steps, i.e. 70 days ahead. This clearly shows the inter-annual differences in catchment response. In the wet year, the fast runoff, such as overland flow and fast subsurface flow, will be large and reach rivers and channels in a shorter period of time. In the dry year, the amount of fast runoff will be reduced due to the large fraction of precipitation percolating into deep soil layers. The percolated water is released to stream flow slowly.

The negative weights in Fig. 9 indicate that a fraction of the precipitation is lost, probably due to water storage at surface, such as in reservoir, ponds, lakes, wetlands and flooded paddy fields, which are abundant in our study area. The stored water will come back again to the stream flow later on, likewise the stream flow mechanism in South America (Vörösmarty et al., 1989; Vörösmarty et al., 1996), where the large inundated forest area release surface runoff gradually.

### 4.3 Simulation with the second implementation of the discrete rainfall-runoff model

When taking the water balance at the surface into account, the WSS and inundated area determine the partition of precipitation into overland flow and infiltrated flow at each time step (Eq. (8)). The simulated stream flow using the second implementation with the duration of 1 time step (Fig. 4) is similar with the simulation using the first implementation (Fig. 3). When the duration of antecedent precipitation was increased in both years, however, the simulated stream flow using the 2[nd] implementation gets closer to the observations (Fig. 4). The changes of NSE and RRMSE (Fig. 6 and 7) also prove this





phenomenon. With the second implementation, a similar model performance is obtained with a duration of 8 time steps to the one with the first implementation and a duration of 15 time steps in 2002 (Fig. 6). In 2005, a similar result as the first implementation and a duration of 15 time steps was obtained with the second implementation, but a duration of 7 time steps (Fig. 7). When the duration of antecedent precipitation was 15 time steps, the simulated stream flow almost overlaps with the

observations in both years (Fig. (4)), with the NSE = 0.98 and RRMSE = 9.2 % respectively in 2002 (Fig. 6), and 0.99 and 3.7 % respectively in 2005 (Fig. 7). This model performance is very good according to the statistical assessment of hydrologic model performances by (Ritter and Muñoz-Carpena, 2013).

The increased duration of antecedent precipitation also largely reduces the model bias from January to March and after June, and the RRMSE from April to June is always lower than 10 % (Fig. 8b). This strongly indicates that the surface wetness

conditions, i.e. the WSS and inundated area, play an essential role in stream flow production in this study area, both in wet and dry years. To reach a required model accuracy, the duration of precipitation in the second implementation can be shorter than that in the first one, mainly because the WSS and inundated area account for the effect of older precipitation, thus the weights of precipitation can be better estimated by Eq. (9) than by Eq. (4).

The weights of overland flow and infiltrated flow do not have as clear a pattern as the weights of precipitation, thus their

distribution is illustrated by a box plot (Fig. 10). The positive values in the weights of overland and infiltrated flows mean that a fraction of component flows does contribute to the stream flow, while the negative values mean that a fraction of overland flow is stored in soil, aquifers and natural or artificial reservoirs, like ponds and lakes, or is consumed by vegetation and human activities. The weights of component flows (Fig. 10) can help to interpret how the precipitation contributes to stream flow as the weights shown in Fig. 9. For example, the precipitation has large weights at antecedent time steps of 1

and 2 in 2002 (Fig. 9a) and of 7 in 2005 (Fig. 9b). At the same time steps, the weights have positive mean values for both overland and infiltrated flows (Fig. 10). This means that the large weight of precipitation is due to overland and infiltrated flows, which both contribute to stream flow at that time step. For the infiltrated flow at the antecedent time steps of 1 and 2 in 2002, the large weights are probably due to the fast subsurface flow, while at longer antecedent steps, such as of 7 in 2005, the large weights of the infiltrated flow are mainly due to the slow subsurface flow. At antecedent time step of 1 in

2005, the weights have positive mean values for both overland (Fig. 10c) and infiltrated flow (Fig. 10d), while the precipitation contribution weight at this time step (Fig. 9b) is relatively low. This difference may explain why the second implementation performs better than the first implementation.

The temporal pattern of weights in Fig. 10 shows clear positive-negative changes for both overland and infiltrated flow. The positive values indicate that the component flow is released to rivers and channels. The negative values indicate the storage

of component flow such as soil water and ground water, or water in lakes, ponds and reservoir, or water consumed by vegetation and humans. Thus changes from positive to negative values indicate transition from release to storage. Besides the values of weight parameters, the difference in the release-storage transition is also clear between dry and wet years. For example, in the dry year (2005), the time when water storage and consumptive water use is dominant on overland flow, arrives much earlier, i.e. at the third time step (in Fig. 10c), than in the wet year, i.e. at the sixth time step (Fig. 10a). The





duration of negative weights for overland flow is also longer in a dry year than in a wet year, i.e. 3 contiguous time steps in 2005 (Fig. 10c) while only 1 time step in 2002 (Fig. 10a). The dry condition accelerates the transition and makes the water-storage-consumption time longer.

The parameter pattern of overland flow (Fig. 10a and 10c) strongly indicates that not all of the produced overland flow

5    immediately reaches rives and channels in our study area. This is also reflected in distributed hydrological models. For example, in TOPMODEL (Beven et al., 1984), the velocity of surface runoff is adapted to calculate the concentration time of overland flow. The human impacts on overland flow are also significant in our case, although many current hydrological models face the problem to fully describe the human impacts on seasonal variations of stream flow and the water cycle in a catchment.

### 4.4 Simulation with the third implementation form of the discrete rainfall-runoff model

In the third implementation of the discrete rainfall-runoff model, we use the field-measured ground water table depth to estimate the potential subsurface flow from the infiltrated flow (Eq. 10). The simulated stream flow using the third implementation with the duration of 1 time step (Fig. 5) is similar with those obtained with the other two implementations

(Fig. 3 and 4). When the duration of antecedent precipitation increases, the simulated stream flow using the third implementation (Fig. 5) becomes close to the observation as fast as the second implementation (Fig. 4). This is also clear in the model performance of 2002 (Fig. 6) and 2005 (Fig. 7).With the same duration, the model performance of the third implementation is almost the best among the three implementations in both years (Fig. 6 and Fig. 7). With the duration of 15 time steps, the NSE = 0.99 and RRMSE = 2.5 % in 2002 (Fig. 6), and 0.99 and 4.2 % respectively in 2005 (Fig. 7). This

good model performance can prove that the potential subsurface flow can be estimated through Eq. (10).

The stream flow in 2005, modelled with the third implementation in 2005 (Fig. 5b), was almost the same as the one modelled with the 2[nd] implementation and various antecedent durations (Fig. 4b). The model performance with the second and third implementation in Fig. 7 was almost identical. Moreover, the weight parameter pattern of the potential subsurface flow in the third implementation in 2005 (Fig. 11d) are very similar with that of the infiltrated flow in the second

implementation (Fig. 10d) in 2005. This similarity means that the potential subsurface flow contributes to the stream flow at the same time step as the infiltrated flow in dry condition. It is mainly due to that in dry condition, the depth of water saturated subsurface is close to the ground water.

The significant changes of RRMSE in each season for both the second (Fig. 8b) and the third implementation (Fig. 8c) indicate the possible period for ground water recharge in our study area. In the duration range between 8 or 9 and 15 time

steps, the RRMSE in each season improved slightly, while with the durations shorter than 8 or 9 time steps, RRMSE values were much higher in each season. That is because with the durations longer than 8 or 9 time steps, antecedent precipitation mainly contributes to aquifer recharge and produces base flow, rather than stream flow. In the second and third implementations, the field-measurements of ground water table depth are used to calculate base flow, thus increasing the



durations of antecedent precipitation beyond 8 or 9 time steps will not reduce the RRMSE as much as as increasing the durations of antecedent precipitation up to 8 or 9 time steps. With the duration of 9 time steps in the third implementation, NSE = 0.98 and RRMSE = 11 % in 2002 (Fig. 6), and 0.97 and 11 % respectively in 2005 (Fig. 7). This model performance is good enough to simulate stream flow.

## 4. 5 Model validation

For each implementation, we calibrated the weight parameters with different durations of antecedent precipitation in 2002 and 2005. To evaluate model performance without calibration, these parameter values were used to estimate the stream flow in 2001 in three experiments: 1) parameters calibrated with the observed stream flow in 2002: 2) parameters calibrated with
10 the observed stream flow in 2005; 3) parameter averages over the 2002 and 2005 calibrations. The best NSE and RRMSE in each implementation (Table 1) can help us to evaluate the model performance, and the results on model validation are summarized as follows:

1)  First implementation. The model performance changes with both experiment types and durations of antecedent precipitation (Fig. 12a). With the same duration, the experiment 3 gives the highest NSE ($0.69 \leq \text{NSE} \leq 0.82$)
and lowest RRMSE ($35.0\% \leq \text{RRMSE} \leq 58.5\%$) among all experiments. NSE for the other two experiments fluctuates in a similar way but in a lower range and RRMSE in the higher range. In the experiment 3, when the duration $\leq 2$ time steps, or $7 \leq$ duration $\leq 11$ time steps, model performance is better than with other durations. The highest NSE and lowest RRMSE both occur in the experiment 3, but the former with duration = 9 time steps and the latter with duration = 2 time steps (Table 1).

2)  Second implementation. The influence of experiment type and durations of antecedent precipitation on model performance is complicated. There are three stages in NSE (Fig. 12b): 1) when the duration $\leq 5$ time steps, the experiment 1 and experiment 3 have a similar NSE ($0.74 \leq \text{NSE} \leq 0.86$), which is better than the experiment 2; 2) when $5 <$ duration $\leq 10$ time steps, the experiment 3 has the highest NSE ($0.63 \leq \text{NSE} \leq 0.80$); 3) when the duration $> 10$ time steps, NSE for all experiments is lower than 0.5. Performance evaluated with RRMSE was
similar, but with different duration ranges (Fig. 12b): 1) when the duration $\leq 3$ time steps, $30.0\% \leq \text{RRMSE} \leq 48.3\%$ for experiment 3 and was close to that for experiment 1, which is better than for experiment 2; 2) when $3 <$ duration $\leq 8$ time steps, the experiment 3 gave the lowest RRMSE ($29.8\% \leq \text{RRMSE} \leq 47.6\%$;); 3) when the duration $> 8$, RRMSE for all experiments was higher than 50 %. The highest NSE occurred in the experiment 1 with duration = 1 time step, while the lowest RRMSE occurred in the experiment 3 with duration
= 8 time steps (Table 1). When the duration $> 10$ time steps, both NSE and RRMSE are not acceptable.

3)  Third implementation. The influence of experiment type and durations of antecedent precipitation on model performance is also complicated. The model performance is acceptable only when duration $\leq 10$ time steps (Fig. 12c). During this period, the experiment 3 has the highest NSE ($0.69 \leq \text{NSE} \leq 0.85$) and lowest RRMSE





(37.9 % ≤ RRMSE ≤ 54.8 % ). The highest NSE and lowest RRMSE both occur in the experiment 3 with the duration = 1 time step (Table 1).

We used the parameters with the best model performance in each implementation (Table 1) to model the stream flow in 2001 (Fig. 13). The modelled stream flows are similar most of the time. During the rainy season, i.e. $100 \leq$ DOY $\leq 200$, the second implementation gave the best results in the three experiments (Fig. 13), as shown by its small RRMSE in Table 1. With the first implementation, the increasing duration changes the NSE and RRMSE within a smaller range than when applying the second and third implementations. Moreover, the weight pattern obtained with the first implementation, in terms of the timing of maximum weight, is similar for different durations for each year (Fig. 9), while for the second and third implementation, the weight pattern is strongly influenced by the setting of the durations of antecedent precipitation (Fig. 10 and 11). This means that the second and the third implementations are more sensitive to the antecedent durations than the first implementation.

The averaged parameters gave a satisfactory performance for most durations of antecedent precipitation in each implementation, since they are based on the information for both the wet and the dry year. The parameters from 2002 gave a slightly better model performance than those from 2005, with duration $< 7$ time steps in the three implementations, due to the relative wet conditions in 2001. The best model performance was observed in two duration ranges: 1) duration $\leq 2$ time steps; 2) 8 time steps $\leq$ duration $\leq 9$ time steps. The first range is due to the large positive weights of component flows in antecedent time steps $\leq 2$ (Fig. 10 and Fig. 11). The second range is due to the recharge of ground water table, which is estimated as 9 time steps according to Fig. 8b and 8c. By taking the ground water recharge (i.e. increasing the duration to 9 time steps) into account, the discrete rainfall-runoff model can better estimate the stream flow in our evaluation experiments.

## Conclusions

The retrievals of the Water Saturated Soil (WSS) and inundated area from 37GHz microwave observation (Shang et al., 2014) proved very useful to parameterize the relationship between precipitation and stream flow. To fully and conveniently use this information, we developed a discrete rainfall-runoff model of the water balance at catchment scale. This model calculates the stream flow as the weighted sum of precipitations at current and antecedent time steps. To simplify the model calibration, base flow estimated from observed ground water table depths, was added into the discrete rainfall-runoff model. This is the first implementation of our model. Considering the water balance at the surface, the retrievals of the WSS and inundated area were used to partition the precipitation into overland and infiltrated flows. In the second implementation of our model, the two component flows replace the precipitation in the first implementation. The potential subsurface flow can be estimated from the infiltrated flow as a function of the ground water table depth. The third implementation is derived by replacing the infiltrated flow in the second implementation with the potential subsurface flow.



The performance of these three implementations of the discrete rainfall-runoff model highly depends on the duration of antecedent precipitation, which is the period during which precipitation is assumed to contribute to current stream flow. With increased durations from 1 to 15 time steps and calibrated with 10-day average stream flow at Changsha station, the NSE of the first implementation increased from 0.78 to 0.91 in 2002, from 0.82 to 0.94 in 2005, and RRMSE decreased from 34 %

to 22 % in 2002, from 28 % to 17 % in 2005; NSE of the second implementation increased from 0.78 to 0.98 in 2002, from 0.83 to 0.99 in 2005, and RRMSE decreased from 34 % to 9.2 % in 2002, from 27 % to 3.7 % in 2005; NSE of the third implementation increased from 0.86 to 0.99 in 2002, from 0.83 to 0.99 in 2005, and RRMSE decreased from 27 % to 2.5 % in 2002, from 27 % to 4.2 % in 2005. This model performance is very good according to the ranking scheme proposed by Ritter and Muñoz-Carpena (2013).The model performance changes led us to conclude that it is beneficial to include the WSS

and inundated area in hydrological modelling studies.

The pattern of the weight parameters in the first implementation explains the annual difference in the catchment response of the study area between dry and wet years. The pattern of the weight parameters in the second implementation illustrates the release-storage-transition of overland flow and infiltrated flow in the study area. In a dry year, the duration of the period of time when water storage and consumptive water use occur is longer than in a wet year. The weight of potential subsurface

flow has a very similar pattern with the infiltrated flow in the dry year. The changes of the RRMSE of the first implementation in each season demonstrate the influence of rain in spring on stream flow (Lin and Lu, 1991; Zhou, 1991; Zong and Chen, 2000). The changes of RRMSE of the second and third implementations in each season give an estimation of the periods when ground water recharge occurs, i.e. 90 days.

For each implementation, we calibrated the weight parameters with different durations of antecedent precipitation in 2002

and 2005. To evaluate model performance with independent observations, these parameters were used to estimate the stream flow in 2001 in three Experiments: 1) calibration with observed stream flow in 2002: 2) calibration with observed stream flow in 2005; 3) parameter averages over 2002 and 2005. The averaged parameters, i.e. based on both 2002 (wet year) and 2005 (dry year) (Fig. 12), gave a satisfactory model performance. The duration of antecedent precipitation corresponding with the best model performance identifies the mechanisms determining stream flow in the study area.

Overall, with the WSS and inundated area as an essential parameter, the three implementations of the discrete rainfall-runoff model not only model stream flow, but also characterize the catchment response to precipitation and the mechanisms determining stream flow.

**Acknowledge:**

This paper is supported by the Ministry of Science and Technology of the People's Republic of China (Grant no. 2015CB953702), the SFEAR's Recruitment Program of Foreign Experts (also called as 1000 Talent Plan, Grant no. WQ20141100224), and the EU-FP7 CEOP-AEGIS project (Grant no. 212921). This investigation is a cooperative effort of



the Delft University of Technology, Delft, The Netherlands and the State Key laboratory of Remote Sensing Sciences, Institute of Remote Sensing and Digital Earth, Chinese Academy of Sciences, Beijing, China. We would like to thank our data providers: Dr. Weiguo Jiang and Dr. Zhigang Liu of Beijing Normal University, Pro. Zhenghui Xie from the Institute of Atmospheric Physics, Chinese Academy of Sciences, and Ming Yin from the Institute of Geo-Environmental Monitoring, China. Without their support, we cannot realize this work.



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





**Table 1, the best model performance and its type of parameter values for the validation of each implementation of the discrete rainfall-runoff model.**

| | | NSE | RRMSE | Used Parameter | Duration |
|---|---|---|---|---|---|
| The first Implementation | Highest NSE | 0.83 | 35.40% | average | 9 time steps |
| | Lowest RRMSE | 0.82 | 34.10% | average | 2 time steps |
| The second Implementation | Highest NSE | 0.86 | 30.00% | 2002 | 1 time step |
| | Lowest RRMSE | 0.80 | 29.80% | average | 8 time steps |
| The third Implementation | Highest NSE | 0.85 | 37.90% | average | 1 time step |
| | Lowest RRMSE | 0.85 | 37.90% | average | 1 time step |



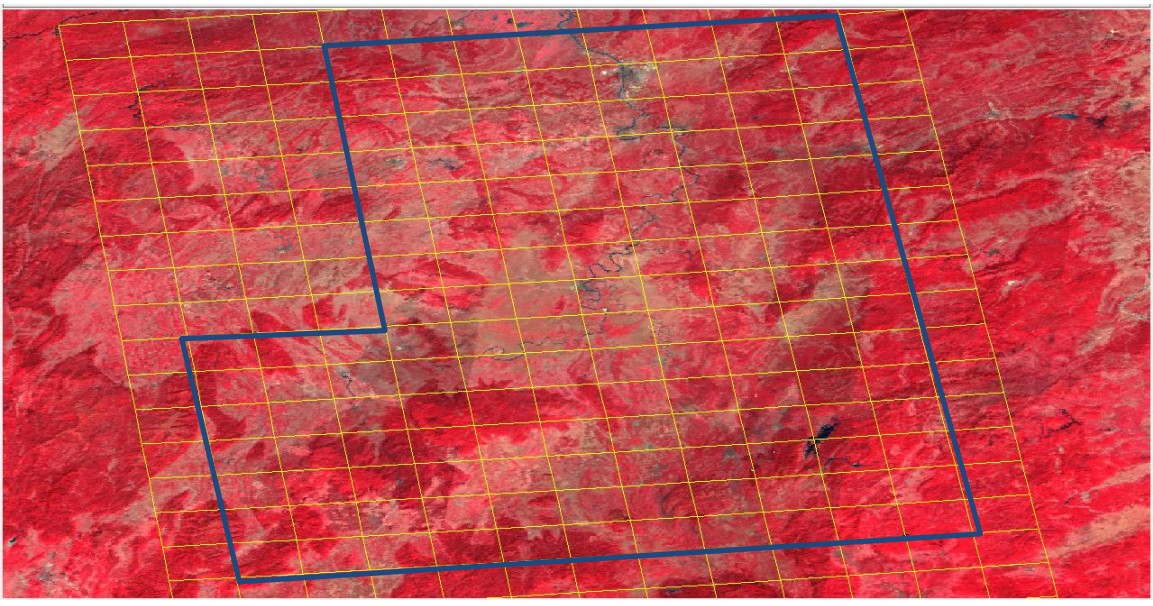

**Figure 1: Study area: the middle and upstream of Xiangjiang river basin. The blue polygon is the study area and the yellow grid is the 25km×25km EASE-Grid; background image: RGB composite of MOIDS near-infrared, red and green spectral bands.**





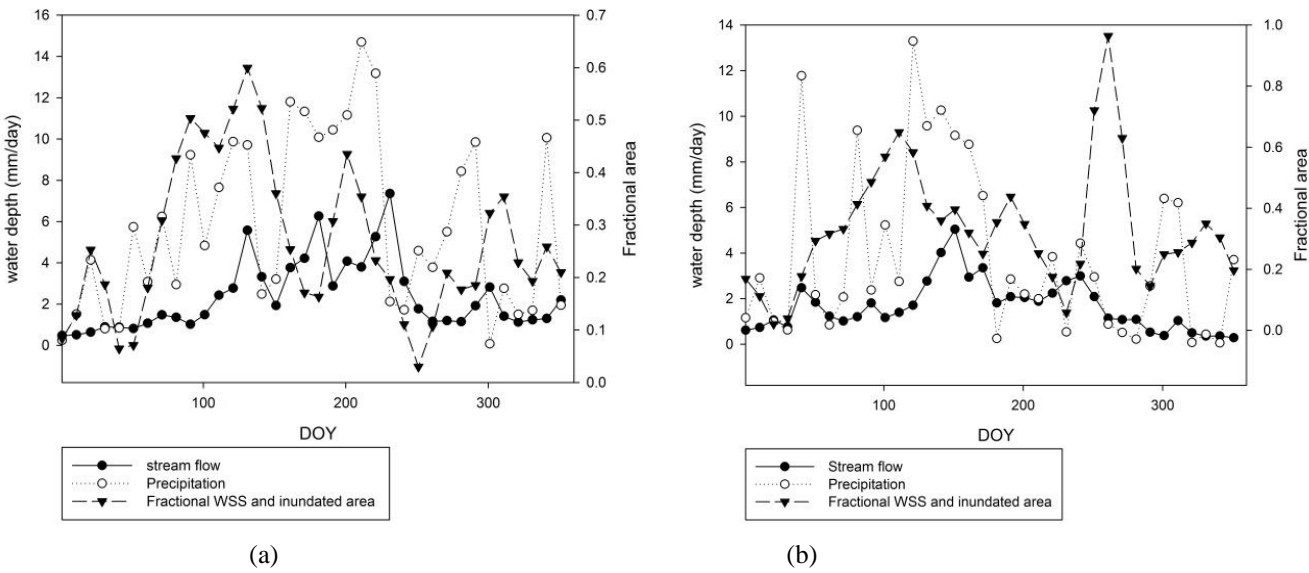

(a)                                    (b)

**Figure 2: Precipitation (10-day averages) extracted from ITP forcing data set (Chen et al., 2011; Yang et al., 2010), fractional WSS and inundated area, and the 10-day averaged stream flow observed at Changsha station, in 2002 (a) and 2005 (b), respectively.**




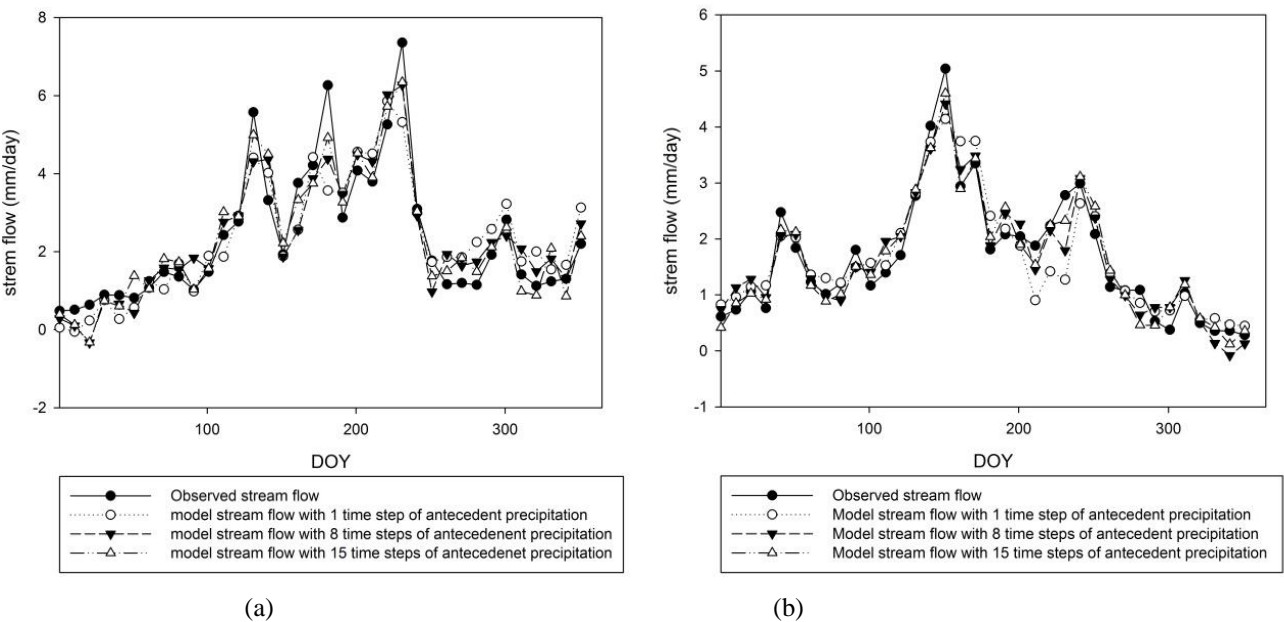

(a)                                                         (b)

**Figure 3: Observed and model stream flow using the first implementation of the discrete rainfall-runoff model with the duration of antecedent precipitation of 1, 8 and 15 time steps in 2002 (a) and 2005 (b) respectively.**




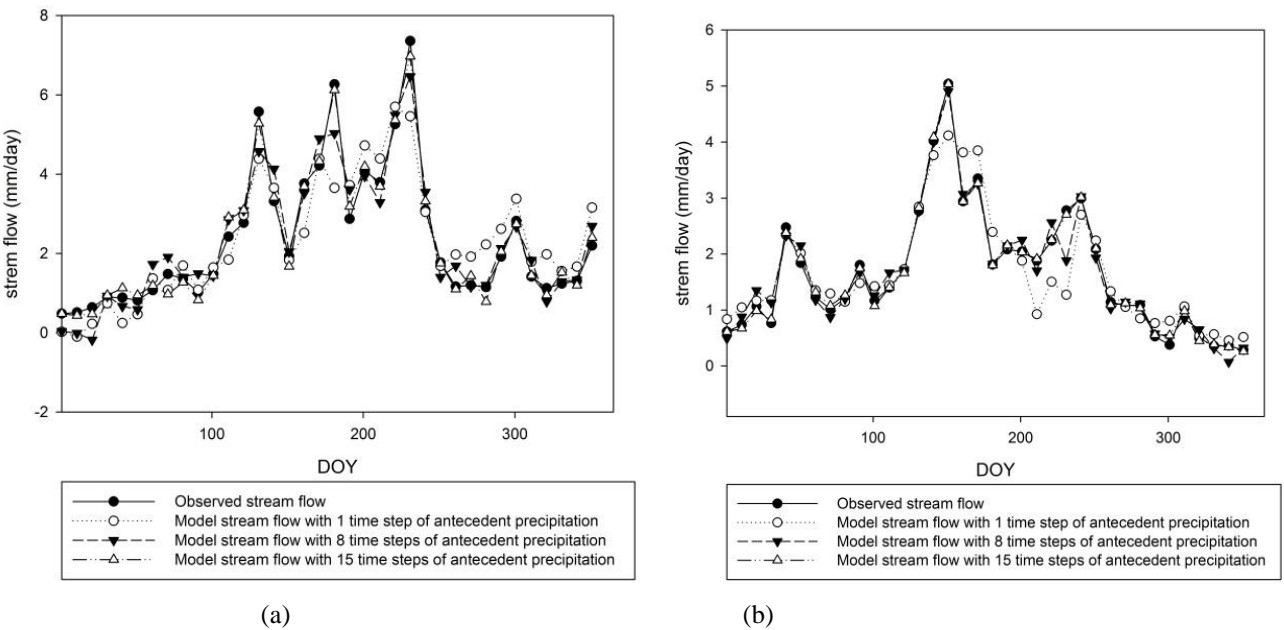

(a)                                                  (b)

**Figure 4: the observed and model stream flow using the second implementation of the discrete rainfall-runoff model with the durations of antecedent precipitation of 1, 8 and 15 time steps in 2002 (a) and 2005 (b) respectively.**





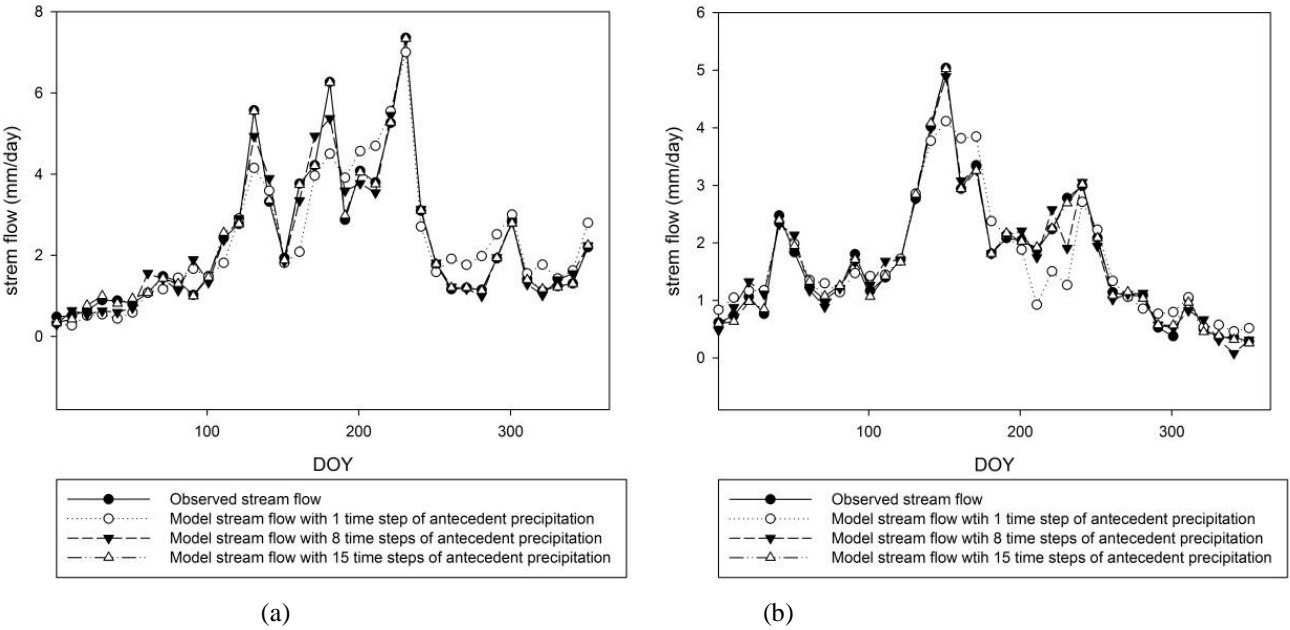

(a)                                             (b)

**Figure 5: the observed and model stream flow using the third implementation of the discrete rainfall-runoff model with the duration of antecedent precipitation of 1, 8 and 15 time steps in 2002 (a) and 2005 (b) respectively.**





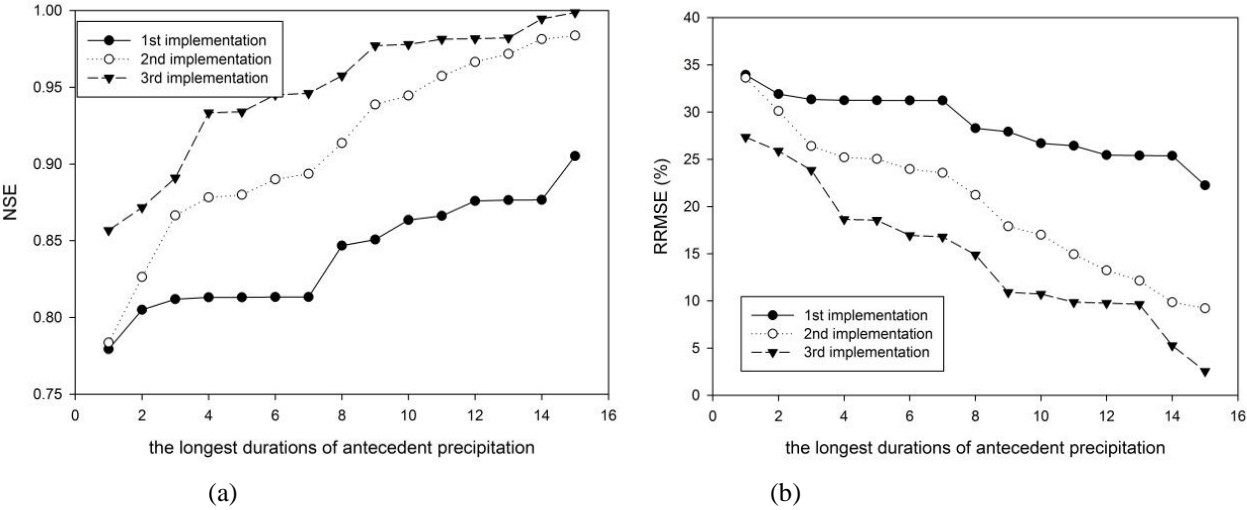

(a)                                                (b)

**Figure 6: the NSE (a) and RRMSE (b) of three implementations of the discrete rainfall-runoff model with the durations of antecedent precipitation from 1 to 15 time steps in 2002: the first implementation with precipitation and base flow; the second implementation with overland flow, infiltrated flow and base flow; the third implementation with overland flow, potential subsurface flow and base flow**





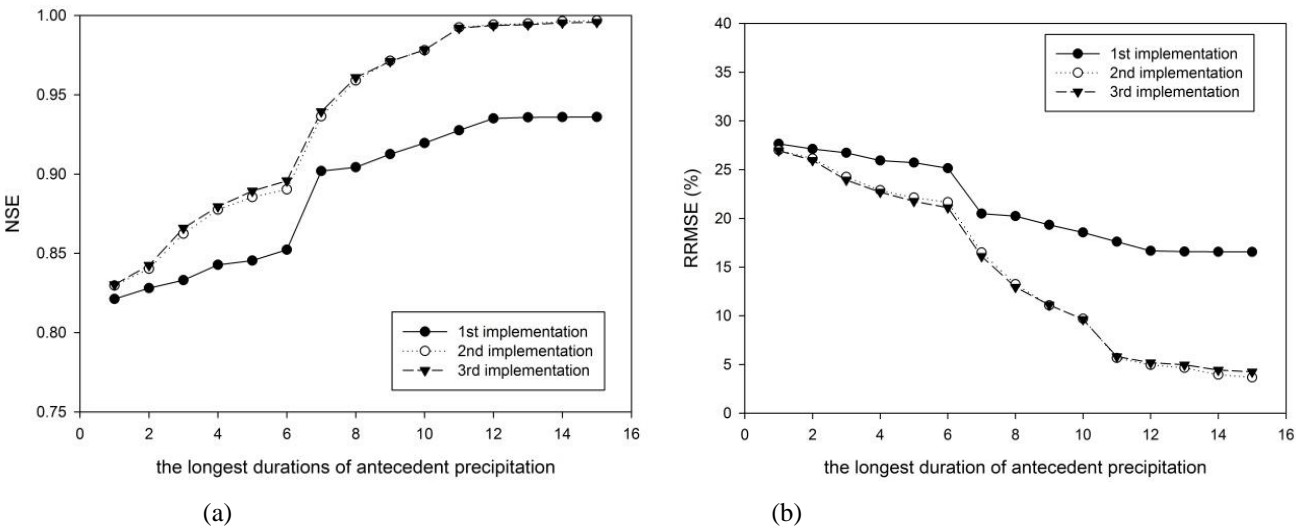

(a)                                          (b)

**Figure 7: the NSE (a) and RRMSE (b) of three implementations of the discrete rainfall-runoff model with the durations of antecedent precipitation from 1 to 15 time steps in 2005: the first implementation with precipitation and base flow; the second implementation with overland flow, infiltrated flow and base flow; the third implementation with overland flow, potential subsurface flow and base flow.**




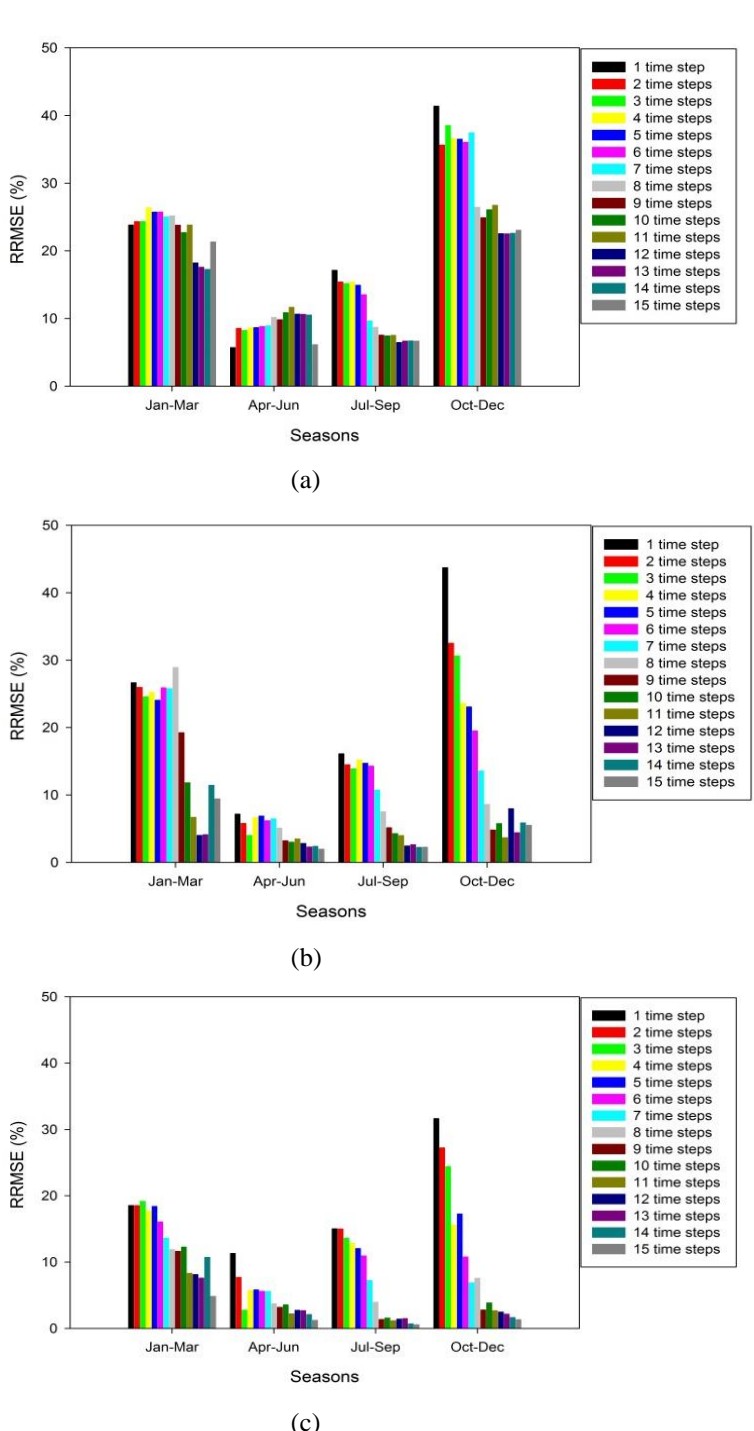

(a)

(b)

(c)

**Figure 8: RRMSE of model stream flow in each season ( in 2002 and 2005) with the duration of antecedent precipitation from 1 to 15 time steps for each implementation of the discrete rainfall-runoff model: (a) the first implementation; (b) the second implementation; (3) the third implementation.**





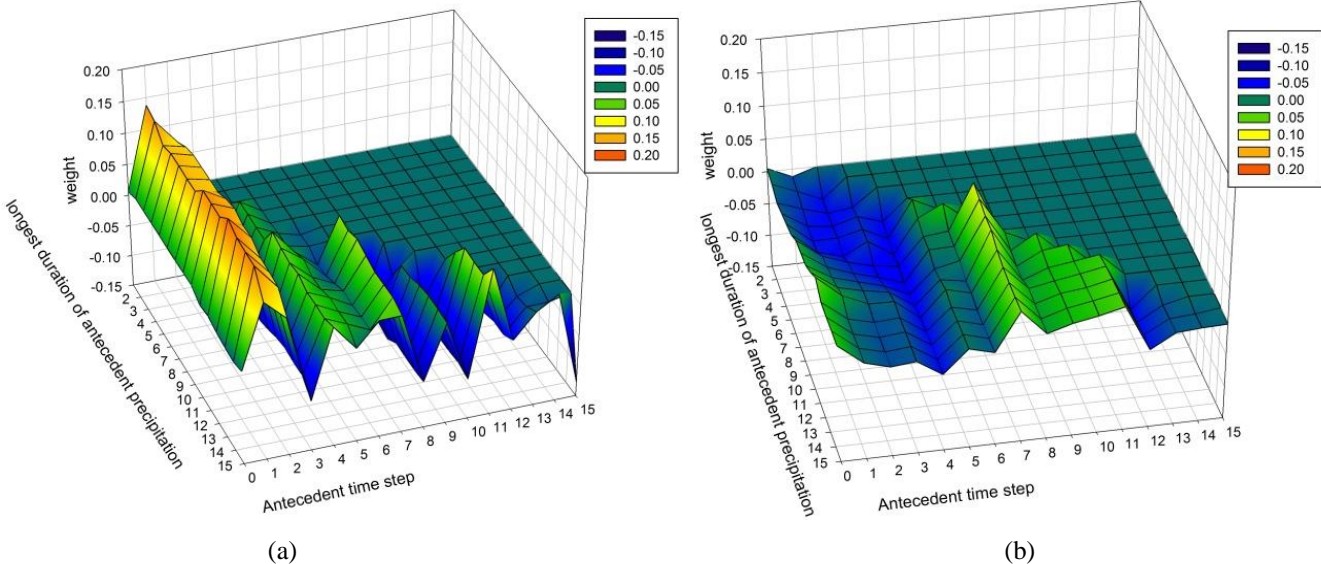

(a)                                                         (b)

**Figure 9: 3D mesh plot of weights of precipitations at each antecedent time step in the first implementation of the discrete rainfall-runoff model with the duration of antecedent precipitation from 1 to 15 time steps: (a) precipitation weight in 2002; (b) precipitation weight in 2005.**





(a)                                                          (b)

(c)                                                          (d)

5  **Figure 10: Box plot of weights of overland and infiltrated flow at each antecedent time step with the second implementation of the discrete rainfall-runoff model: (a) the weights of overland flow in 2002; (b) the weights of infiltrated flow in 2002; (c) the weights of overland flow in 2005; (d) the weights are of infiltrated flow in 2005.**





(a)

(b)

(c)

(d)

5   **Figure 11: Box plot of weights of overland and infiltrated flow at each antecedent time step with the third implementation of the discrete rainfall-runoff model: (a) the weights of overland flow in 2002; (b) the weights of potential subsurface flow in 2002; (c) the weights of overland flow in 2005; (d) the weights are of potential subsurface flow in 2005.**





(a)

(b)

(c)

**Figure 12: NSE and RRMSE for three implementations using parameter values of three experiments to validate model peformance in 2001: a) for the first implementation; b) for the second implementation; c) for the third implementation.**







**Figure 13: model stream flow in 2001 using the three implementations of the discrete rainfall-runoff model. The parameters for each implementation are derived according to the best model performance in Table 1.**