# Peer review of "Modelling stream flow with a discrete rainfall-runoff model and 37GHz PDBT microwave observations: the Xiangjiang River basin case study"

_Hydrology and Earth System Sciences, 2016_

## Referee Comment (RC1) · Anonymous Referee #1 · 10 Aug 2016

In this paper, the authors develop the new rainfall-runoff model. They introduce the newly developed inundation data product into the semi-empirical hydrological model. They improve the skill of simulating streamflow using the observed groundwater storage and the inundation data.

Unfortunately, I recommend the editor not to include this discussion paper in the HESS journal. Although the topic of this paper is suitable to HESS, I think the design of this study has several major deficiencies which may not be fixed in a short period of time.

The most important conclusion of this paper is that the retrievals of Water Saturated Soil (WSS) and inundated area improve the model accuracy. However, I cannot agree with it. The model performance should be evaluated in the calibration/validation process. The authors should not conclude that they can improve the accuracy just by the results in the calibration period. Figure 12 clearly shows that there is no improvement from their first implement to second implement in their model validation. Their validation clearly rejects their hypothesis.

The authors may need to pay more attentions to the overfitting issue. For example, the accuracy of their rainfall-runoff model is improved when they increase the longest durations of antecedent precipitation. The authors interpret this result in terms of the basin characteristics. However, when the authors increase the longest duration of antecedent precipitation, the performance of the linear regression should be always improved since the number of weights (unknown parameters) increases. It is possible that the improvement of the rainfall-runoff model in this paper is not related to any real rainfall-runoff processes. The authors may need to check if their statistical model does not overfit to the data using the cross-validation analysis. In the second and third implementation, the model has better performances with the larger durations of antecedent precipitation in the calibration period (Figures 6 and 7) while their performances are degraded by increasing the duration in the validation period (Figure 12). In the case of second and third models, increasing the duration causes the overfit to the data.

In addition, I believe that the number of unknown parameters also increases when the authors upgrade their model from the first to second implementation. The authors split the weight (w) into two other parameters (beta). It is possible that the improvement of the model is just caused by increasing the number of parameters and overfitting to the data. In validation, the second and third implementations cannot significantly outperform the first implementation. It indicates the second and third implementations overfit to the calibration data. I am not convinced that the WSS and inundated area data positively impact to the model performance.

I recommend that the authors to use the model which let us to interpret the results more easily. The physically-based hydrological models is more appropriate since they do not need a heavy calibration.

I give my specific comments below. I hope it helps.

Major Points: Page 2, Line 1: I think this Introduction can be significantly improved. In the Introduction section, the authors should explain their problem to be solved and review the previous works related to this problem. Then, the authors should briefly summarize the increment of their research from the previous papers. The author should not include the detail of their methodology in the Introduction section. I believe that in this paper, both previous contributions and the author's original approaches coexist in the Introduction. I recommend the authors to move some paragraphs to the following method section and focus more on citing the previous works and explaining the context of their problem.

Page 5, Line 1-23: The authors may move these paragraphs to the following method section.

Page 5, Line 17-22: I think the upgrade from the second to third implementation is not related to the use of satellite WSS data. Is the third implementation really needed considering the scope of this paper?

Page 6, Line 16: Is the "discrete" hydrological model identical to the "distributed" hydrological model? Please define the "discrete" model.

Page 10, Line 16-17: Although I understand that their WSS product has been explained in Shang et al. [2014], please explain some details of their product in this paper. I believe that this information is helpful to interpret the author's results. For example, how is the spatial resolution of the product? Is their 37GHz product affected by atmospheric vapor and clouds? In addition, there are some contradictions between precipitation time series and WSS time series in Figure 2 (e.g., sharp peak in DOY 250 of 2005 with no strong rainfall). Please clarify the capability and limitation of their data.

Page 11, Line 29-30: As described above, whenever the number of weights (unknown parameters) increases, the linear regression performs better. I believe that the improvement should not be interpreted in this way unless the authors confirm that the improvement is robust by the calibration/validation process shown in section 4.5.

Page 12, Line 23-24: It is hard for me to interpret the negative weights. The authors explain that negative weights are caused by water storages. It looks a reasonable explanation. However, if the weights are negative, streamflow decreases with increase of rainfall. How can the authors explain this? Do the authors have some previous works which interpret the negative weights in this way? Do the authors think they need a constraint of non-negative weights in the linear regression?

Page 13, Line 6-7: Please explain this statistical assessment more deeply and discuss the model performance quantitatively.

Page 13, Line 8-11: As discussed above, this interpretation is valid after the authors perform the validation and confirm the robustness of the model's improvement.

Page 13, Line 32-35: At the third time step, the weight goes negative. However, it goes positive at the sixth time step again. How can the authors interpret it?

Page 14, Line 32-33: I believe that the authors also use ground water table depth data in the first implementation. Please clarify it.

Page 15, Line 6: As discussed, I strongly believe that this model validation rejects the authors' hypothesis.

Minor Points: Page 7, Line 13 - Page 8 Line20: I recommend the authors to move this sub-sections to the end of this method section. The model calibration and the evaluation metrics are applied to all of implementations so that it should not be included only in the first implementation section. Page 9, Line 32: Beta 21 and Beta22 may be Beta31 and Beta32, respectively. Page 10, Line 15: Please show the locations of 9 wells in the river basin. Page 14, Line 5: rives→rivers

---

## Referee Comment (RC2) · Anonymous Referee #2 · 12 Aug 2016

The authors have developed a discrete rainfall-runoff model which uses ground data as well as retrievals of Water Saturated Soil (WSS) and inundation area from 37GHz microwave observations. I had a difficulty in understanding the objective, finding, and the contribution of this manuscript. There are many questions which were not addressed by authors clearly, for example.

1. Use of 37 Ghz (which sensor/satellite?) for WSS and inundation estimation why not other frequencies, which are commonly used for soil moisture and inundation area estimation?

2. What is discrete rainfall-runoff model, why it is better than other models?

[Figure]

3. Why the model was used at a time step of 10 days? Under which circumstances this approach is valid?

4. What is the purpose of 3 step implementation? Where do get the evapotranspiration data from? What happened to this approach if we do not enough data from the ground (e.g water table) 5. What is the role of calibration in the model results?

6. What is the purpose of this manuscript? Model development? Or use of satellite observation for improving model simulation?

In addition, manuscript was not well written and arranged . I recommend authors to re-structure the paper in a way that readers can understand the concept, their applications to solve the practical issues using your model or approach.

Specific issues are listed here for future modification.

Page 2: Line 1-5: Introduction does not explain the advantages and short coming of 37 GHz for estimating soil wetness and inundation area. I do think that use of 37 GHz for the estimation of soil wetness as well as inundation area is not the right choice since it is affected by clouds and water vapor and thus affecting the PDBT significantly. I believe the authors may know much better about the soil moisture estimation from low frequency microwave observation (1, 6.9 and 10 GHz). Provide reasons why 37 GHz was used rather than 6.9 and 10 GHz. I could not able to understand why does the discrete rainfall-runoff modeling approach better than other available or physical model approaches and what are the advantages?. If you use 36 GHz for inundation why not use 89 GHz which has high resolution than 36 GHz.

Page 2-5: Mixer of everything very difficult to follow. Better to improve the introduction and move equations and description of model to Section:Method, especially page 5.

Page4: Equation 3: Confusing. Weight w is divided by P and again multiplied by P? Please simplify and explain clearly.

Page 10: Line 10-25. What happened to evapotranspiration? What data was used to

calculate evapotranspiration to solve eq. 3?

Page 11: why does the model have the time steps of 10 days? What is the advantage of this approach? Can you use this approach for simulating peak discharge?

Page 11-15: what is the target this manuscript? Reproduction of stream flow? In that case why other available physical model cannot be used? How to understand that WSS and inundation area really improved the model performance?

I could not able to follow the results and discussion completely since the previous sections could not explain clearly.

---

## Author Comment (AC1) · 10 Dec 2016

The comment was uploaded in the form of a supplement:
http://www.hydrol-earth-syst-sci-discuss.net/hess-2016-129/hess-2016-129-AC1-supplement.zip

---

## Author Comment (AC2) · 10 Dec 2016

The comment was uploaded in the form of a supplement:
http://www.hydrol-earth-syst-sci-discuss.net/hess-2016-129/hess-2016-129-AC2-supplement.zip